# SQS: Enhancing Sparse Perception Models via Query-based Splatting in Autonomous Driving

**Haiming Zhang**[1,2]\*, **Yiyao Zhu**[3]\*, **Wending Zhou**[1,2], **Xu Yan**[4][✉],
**Yingjie Cai**[4], **Bingbing Liu**[4], **Shuguang Cui**[2,1], **Zhen Li**[2,1][✉]

[1] FNii, Shenzhen  [2] SSE, CUHK-Shenzhen
[3] HKUST  [4] Huawei Noah's Ark Lab
{haimingzhang@link.,lizhen@}cuhk.edu.cn
yzhucp@connect.ust.hk
yanxu44@huawei.com

## Abstract

Sparse Perception Models (SPMs) adopt a query-driven paradigm that forgoes explicit dense BEV or volumetric construction, enabling highly efficient computation and accelerated inference. In this paper, we introduce SQS, a novel query-based splatting pre-training specifically designed to advance SPMs in autonomous driving. SQS introduces a plug-in module that predicts 3D Gaussian representations from sparse queries during pre-training, leveraging self-supervised splatting to learn fine-grained contextual features through the reconstruction of multi-view images and depth maps. During fine-tuning, the pre-trained Gaussian queries are seamlessly integrated into downstream networks via query interaction mechanisms that explicitly connect pre-trained queries with task-specific queries, effectively accommodating the diverse requirements of occupancy prediction and 3D object detection. Extensive experiments on autonomous driving benchmarks demonstrate that SQS delivers considerable performance gains across multiple query-based 3D perception tasks, notably in occupancy prediction and 3D object detection, outperforming prior state-of-the-art pre-training approaches by a significant margin (*i.e.,* +1.3 mIoU on occupancy prediction and +1.0 NDS on 3D detection).

## 1 Introduction

Recent advances in vision-centric autonomous driving have driven significant progress in the field [7, 60]. From a representation standpoint, existing approaches can be broadly categorized into dense BEV-centric and sparse query-centric paradigms. Dense BEV-centric methods [24, 44, 13] extract Bird's Eye View (BEV) features from multi-view images for downstream tasks, while Sparse Perception Models (SPMs) [55, 33, 31] bypass explicit dense representations and directly aggregate features from images using implicit queries, enabling faster inference. Sparse query-centric methods have garnered increasing attention within the community due to their practical advantages for real-world deployment. Despite the dominance of supervised methods, their reliance on precise ground-truth annotations presents a substantial challenge, as acquiring such labels is both costly and labor-intensive. Conversely, the abundance of unlabeled data offers a promising avenue to further enhance model performance. Nevertheless, effectively leveraging this data remains a non-trivial challenge.

To mitigate these challenges, various pre-training strategies have been proposed for autonomous driving. Earlier works leverage contrastive learning [45] and Masked Autoencoders (MAE) [38] for pre-training. However, the coarse supervision they provide limits their capacity to fully capture spatial-temporal geometry. In contrast, NeRF-based approaches such as UniPAD [62] and ViDAR [64] utilize

---

\*Equal contribution. Work done during internship at Huawei.

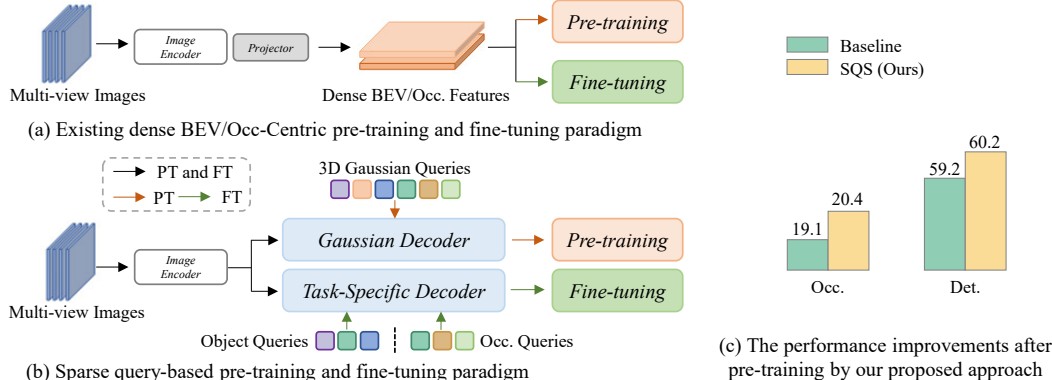

(a) Existing dense BEV/Occ-Centric pre-training and fine-tuning paradigm

(b) Sparse query-based pre-training and fine-tuning paradigm

(c) The performance improvements after pre-training by our proposed approach

Figure 1: **The comparison of pre-training and fine-tuning paradigms.** (a) Existing pre-training approaches operate on dense BEV or Occupancy representations, which are subsequently shared during fine-tuning. (b) The proposed SQS can be integrated into any sparse query-based perception model, accepting Gaussian queries for pre-training and utilizing them for prediction. (c) We demonstrate the effectiveness of SQS on query-based 3D semantic occupancy prediction (Occ.) and 3D object detection (Det.) tasks. PT and FT denote pre-training and fine-tuning, respectively.

3D volumetric differentiable rendering to reconstruct and predict 3D scene structures. To reduce memory overhead and improve rendering efficiency, a separate line of research [58, 69] introduce 3D Gaussian Splatting (3DGS) [19] for explicit scene representation. By predicting Gaussian parameters for feedforward reconstruction, GaussianPretrain [58] achieves comprehensive scene understanding by integrating geometric and texture representations. Moreover, VisionPAD [69] projects neighboring frames onto the current frame using rendered depths and relative poses, relying solely on RGB image supervision rather than explicit depth annotations. Although existing pre-training paradigms have substantially enhanced the performance of downstream applications, their reliance on dense BEV representations limits their applicability to SPMs (Fig. 1(a)).

In this paper, we present the first attempt to pretrain Sparse Perception Models (SPMs) on unlabeled data to enhance their downstream performance, as shown in Fig. 1(b). Unlike dense BEV-centric perception models, we find out that the latent sparse queries in SPMs lack explicit spatial positions and semantic meanings, making it challenging to directly apply existing rendering-based pre-training methods, which often fail to preserve informative representations during training. To address this challenge, we propose SQS, a novel pre-training framework for SPMs based on query-based splatting. Unlike previous approaches, SQS introduces a small set of adaptive Gaussian queries during pre-training. These queries dynamically predict 3D Gaussians and reconstruct both depth and RGB images via a splatting mechanism, enabling the model to learn fine-grained representations from unlabeled data in a self-supervised manner. After pre-training, the learned Gaussian queries are used for fine-tuning, where they interact and fuse with task-specific queries, resulting in improved downstream performance. We evaluate our approach on tasks such as object detection and 3D occupancy prediction. Experimental results demonstrate that SQS consistently achieves significant performance improvements over state-of-the-art methods without pre-training.

To this end, our contributions can be summarized as follows:

- We propose SQS, the *first* query-based splatting pre-training technique specifically designed to advance Sparse Perception Models (SPMs).

- We introduce plug-and-play Gaussian queries, which learns fine-grained features in a self-supervised manner during pre-training, and further enhances downstream tasks via interactive feature fusion during fine-tuning.

- SQS significantly enhances performance in both occupancy prediction and 3D object detection, surpassing previous state-of-the-art results on multiple autonomous driving benchmarks (*i.e.,* +1.3 mIoU on occupancy prediction and +1.0 NDS on 3D detection as Fig. 1(c)).

## 2  Related Work

**Pre-training in autonomous driving.**  Pre-training has gained remarkable progress in recent years for autonomous driving. Conventional approaches cover supervised [42, 54, 50, 59], contrastive [23, 41, 45, 65], and masked signal modeling [39, 38, 2, 20] categories. With advances in neural rendering [37], rendering-based pre-training [62, 64, 74, 66] becomes an alternative by rendering images from dense BEV or Volume representation. For example, UniPAD [62] utilizes 3D volumetric differentiable rendering to reconstruct 3D shape structures and appearance characteristics. Meanwhile, ViDAR [64] predicts the future point cloud using a Latent Rendering operator based on historical embeddings. To achieve effective and efficient rendering, 3D Gaussian Splatting [19] is introduced in some recent works. GaussianPretrain [58] considers 3D Gaussian anchors as volumetric LiDAR points for unified geometric and texture representations. Without explicit depth supervision, VisionPAD [69] reconstructs images employing both voxel velocity estimation and multi-frame photometric consistency. These pre-training pipelines successfully model spatial-temporal representation for dense BEV features.

However, the emerging perception methods with sparse route [55, 33, 34, 52, 31] are not compatible with the paradigms above. Recently, the query-based pre-training in 2D image has been developed. Frozen-DETR [10] utilizes frozen foundation models with class token and patch token, which provide a compact context and semantic details, respectively. GLID [32] models pre-training pretext task and other downstream tasks as "query-to-answer" problems. Since the sparse pre-training in autonomous driving requires an accurate 3D geometric representation extracted from multi-view images, the existing methods for 2D are inapplicable for Sparse Perception Models (SPMs).

**Sparse Perception Models for 3D Detection and Occupancy Prediction.**  For the sparse 3D detection, motivated by DETR [5], DETR3D [55] utilizes a sparse set of 3D object queries to sample the 2D features from images by 3D point projection. To avoid the complex 2D-to-3D projection and feature sampling, PETR series [33, 34, 52, 18, 30, 67] directly interact with 3D position-aware features by encoding the 3D position into 2D image features. Without relying on dense view transformation nor global attention, Sparse4D [27] iteratively refines anchor boxes via sparsely sampling and fuses spatial-temporal features. In the SparseBEV [31], to adapt the detector in both BEV and image space, a set of sparse pillar queries initialized in BEV space are applied to interact with the image features.

Regarding occupancy prediction, the query-based approaches [51, 48, 47] are proposed to reduce computational cost. OPUS [51] formulates the task as a streamlined set prediction paradigm. SparseOcc [48] proposes an efficient occupancy network with 3D sparse diffuser and convolutional kernels while OSP [47] presents the Points of Interest (PoIs) to represent the scene. Recently, 3DGS has demonstrated the capacity to adapt flexibly to varying object scales and regional complexities in a deformable manner, thereby enhancing resource allocation and overall efficiency. Based on the aforementioned advantages, another line of works utilize 3DGS for supervised [16, 14, 75] or self-supervised [1, 68, 17] occupancy prediction. In conclusion, compared to the BEV based methods, the sparse algorithms reduce computational cost and broaden the perception range. This distinctive advantage makes the development of a sparse pre-training algorithm for them particularly imperative.

**3D Gaussian Splatting in Autonomous Driving.**  3D Gaussian Splatting (3DGS) [19] uses multiple 3D Gaussian primitives for fast radiance field rendering, enabling explicit representation with fewer parameters. For reconstructing driving scenes, several approaches are carefully designed for 3D static [72, 61, 56] and 4D dynamic [63] scenes. More recently, 3DGS based perception models have been proposed, including occ prediction [16, 14, 75], bev segmentaiong [6, 36] and end-to-end tasks [71]. Alternatively, some recent works apply 3DGS for self-supervised occ prediction [1, 17, 68]. GaussianFlowOcc [1] and TT-GaussOcc [68] model scene dynamics by predicting the temporal flow for each Gaussian throughout the training procedure. Without requiring explicit annotations, GaussTR [17] splats the Gaussians onto 2D perspectives and aligns the extracted features with foundation models. Furthermore, regarding to the self-supervised pre-training, sevaral methods [58, 69] adopt 3DGS for explicit geometry representation in the Dense BEV or Volume feature to improve the performance of downstream tasks. Nevertheless, up to now, there is still no pre-training scheme that can effectively adapt to Sparse Perception Models (SPMs).

# 3 Proposed Method

In this section, we introduce our query-based splatting pre-training approach for autonomous driving. The overall architecture of the proposed SQS framework is depicted in Fig. 2.

We first provide the necessary preliminaries on 3D Gaussian Splatting, which enables the rendering of both RGB images and depth maps from predicted 3D Gaussians. Subsequently, we briefly outline the image encoder employed to extract multi-scale features from multi-view input images. We then detail the query-based Gaussian transformer decoder, which utilizes Gaussian queries to predict 3D Gaussians and facilitates the learning of fine-grained information via self-supervised splatting. Finally, by incorporating the pre-trained Gaussian queries through a query interaction module during fine-tuning, our approach effectively transfers knowledge from the pre-training stage, thereby enhancing downstream query-based learning performance.

## 3.1 Preliminaries

3D Gaussian Splatting (3DGS) [19] represents 3D scenes through collections of $K$ Gaussians. Each primitive $g_k$ contains 3D position $\mu_k \in \mathbb{R}^3$, covariance $\mathbf{\Sigma}_k$, opacity $\alpha_k \in [0,1]$, and spherical harmonics coefficients $c_k \in \mathbb{R}^k$.

For differentiable optimization, the covariance matrix is parameterized using scaling $\mathbf{S} \in \mathbb{R}^3_+$ and rotation $\mathbf{R} \in \mathbb{R}^4$ matrices:

$$\Sigma = \mathbf{R}\mathbf{S}\mathbf{S}^T\mathbf{R}^T. \tag{1}$$

Projection to image coordinates employs view transformation $\mathbf{W}$ and Jacobian $\mathbf{J}$:

$$\Sigma' = \mathbf{J}\mathbf{W}\mathbf{\Sigma}\mathbf{W}^T\mathbf{J}^T. \tag{2}$$

Rendering combines ordered Gaussians using an alpha-blend rendering proceduure [37], and color at pixel $p$ is computed as:

$$\mathbf{C}(p) = \sum_{i \in K} c_i \alpha_i \prod_{j=1}^{i-1}(1 - \alpha_i). \tag{3}$$

To introduce geometric representation [8], depth rendering is computed as:

$$\mathbf{D}(p) = \sum_{i \in K} d_i \alpha_i \prod_{j=1}^{i-1}(1 - \alpha_j), \tag{4}$$

where $d_i$ represents the distance from $i$-th Gaussian to the camera. Unlike volume rendering [37], 3DGS uses efficient splat-based rasterization that projects 3D Gaussians as 2D image patches.

## 3.2 Image Encoder

Given a set of multi-view images $\mathcal{I} = \{\mathbf{I}_i \in \mathbb{R}^{3 \times H \times W} | i = 1, ..., N\}$, corresponding intrinsics $\mathcal{K} = \{\mathbf{K}_i \in \mathbb{R}^{3 \times 3} | i = 1, ..., N\}$ and extrinsics $\mathcal{T} = \{\mathbf{T}_i \in \mathbb{R}^{4 \times 4} | i = 1, ..., N\}$ as inputs, where $N$ is the number of cameras. We first need to extract multi-scale multi-view image features for the subsequent decoder. Specifically, we feed multi-view images to the backbone network (e.g., ResNet-101 [11]), and obtain the intermediate multi-level feature $F'$. To further enhance and aggregate these features across different spatial resolutions, we utilize a Feature Pyramid Network (FPN). The FPN processes the multi-level features and produces multi-scale image features $F$, which effectively captures both high-level semantic information and fine-grained spatial details.

## 3.3 Gaussian Transformer Decoder and Gaussian Queries

As illustrated at the top of Fig. 2, SQS employs a Gaussian Transformer Decoder to process 2D image features and reconstruct multi-view RGB and depth images. This reconstruction enables the model to capture the underlying geometry and appearance of Gaussian attributes, providing a strong feature prior. As a result, SQS enhances downstream sparse perception tasks by supplying a pretrained image backbone and enriched Gaussian query representations.

Each Gaussian query is initialized as learnable anchors $g_k \in \mathbb{R}^{K \times C}$, paired with queries $q_k \in \mathbb{R}^{K \times D}$ using zero vectors in high-dimensional space, where $K$ is the number of Gaussians, $C$ and $D$ are

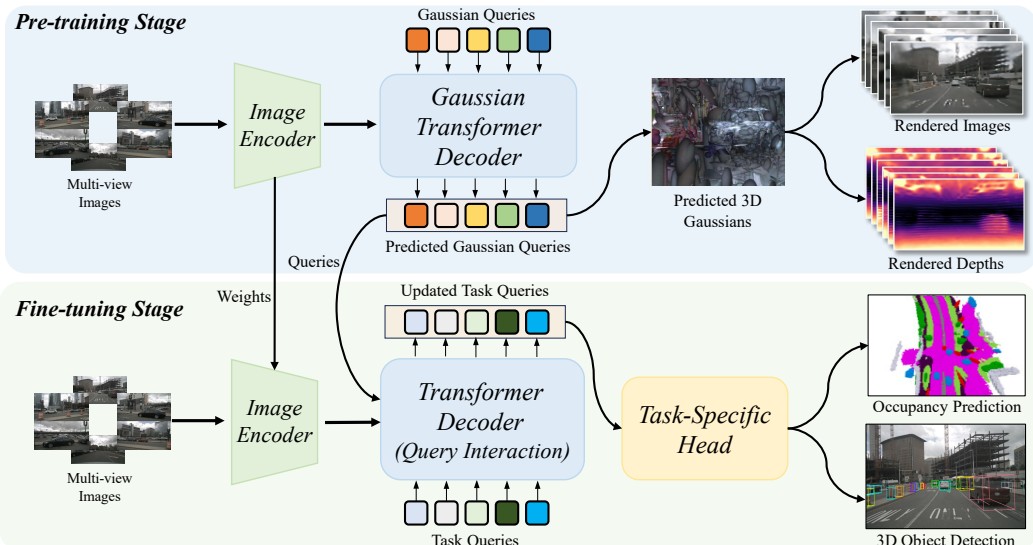

Figure 2: **The pipeline of our proposed SQS.** In order to adapt the sparse query-based downstream tasks, we design a sparse query-based 3D Gaussian Splatting pre-training paradigm with RGB image and depth as supervision. The pre-trained image encoder can be leveraged during the fine-tuning stage, and we also propose a query interaction module to fully exploit the knowledge encapsulated in the pre-trained queries. Our proposed light-weight pre-training paradigm can be plugged into any sparse query-based downstream tasks to enhance their performance.

the dimension of Gaussian primitives and query features respectively. During pre-training, guided by the initialized vectors with learnable Gaussian primitives, these query features interact through self-encoding and deformable cross-attention with image features to predict the Gaussian attributes, enabling the retention of rich and detailed geometric information.

To capture the representation across the entire scene, 3D sparse convolution across Gaussian queries is employed to reduce memory cost with linear computational complexity. Here, the 3D position $\mu_k \in \mathbb{R}^3$ in Gaussian anchor is used to voxelize each Gaussian and the sparse convolution is leveraged on the voxel grid.

Then the multi-level image feature $F$ is aggregated with deformable cross attention. Concretely, for each Gaussian query, multiple 3D reference points are calculated with offsets added to the 3D position $\mu$ from the Gaussian anchor. With the intrinsics $\mathcal{K}$ and extrinsics $\mathcal{T}$, these 3D points are projected onto 2D image planes to facilitate feature sampling. The resulting set of sampled features serves as keys and values in the subsequent attention mechanism.

Finally, to enable the prediction of Gaussian properties, a dedicated Gaussian head comprising multi-layer perceptrons (MLPs) is applied to each Gaussian query. To constrain the predicted parameters within appropriate value ranges, sigmoid activations are applied to the position, scale, and opacity, while the rotation is normalized to unit length. The Gaussian parameters $\mu$, $\alpha$, $c$, $S$ and $R$ are iteratively refined across decoder layers, where only the position $\mu$ is predicted in the form of a delta, while the remaining parameters are directly replaced at each refinement step.

### 3.4 Reconstruction Loss for Pre-training

We employ an L1 loss function for both RGB and depth reconstruction. LiDAR points serve as the ground truth (GT) for depth, and the depth loss is supervised exclusively at pixels corresponding to valid LiDAR measurements. The overall loss is formulated as follows:

$$\mathcal{L} = \omega_1 \mathcal{L}_{\text{rgb}} + \omega_2 \mathcal{L}_{\text{depth}}, \tag{5}$$

where $\omega_1$ and $\omega_2$ are set to 1.0 and 0.05, respectively, to weight the corresponding terms.

### 3.5  Query Interaction for Fine-tuning

Through query-based pre-training, both the image backbone and Gaussian queries acquire rich geometric feature. For fine-tuning downstream SPMs, loading the pre-trained image backbone is straightforward; however, reusing pre-trained Gaussian queries remains challenging. Unlike Dense BEV-Centric perception frameworks, which leverage dense BEV representations as a unified intermediate feature, SPMs typically lack such a common structural basis. Instead, these methods often employ task-specific queries and are paired with dedicated decoders designed for different perception tasks. The decoder architectures across various sparse perception algorithms can differ significantly. For instance, SparseBEV [31] initializes queries in 2D BEV space, whereas PETR [33] conducts initialization in 3D space.

To enable consistent reuse of pre-trained queries across diverse sparse perception tasks, as shown in the bottom of Fig. 2, we propose a plug-in framework based on Query Interaction, facilitating greater flexibility and generalization across diverse tasks. This module explicitly bridges pre-trained Gaussian queries with task-specific queries, facilitating effective transfer and adaptation within heterogeneous decoder frameworks.

Specifically, downstream SPMs initially load the weights of image backbone from a sparse pre-trained model. Regarding the reuse of pre-trained Gaussian queries, we fix the parameters of the lightweight pre-trained model. For each perception case, the pre-trained model infers a set of corresponding Gaussian anchors paired with query features. We set opacity threshold $\alpha_{thresh}$ to filter out anchors with low opacity. To leverage pre-trained queries efficiently, spatial-aware local attention [46] is applied. To elaborate, given 3D position $\mu_t$ of task query $q_t$ and $\mu_k$ from pre-trained anchors $g_k$, we apply $k$-nearest neighbor algorithm to find $k$ closest 3D Gaussians for each task query. To this end, $q_t$ only aggregates features from nearest $k$ Gaussian queries, finally the local query interaction is formulated as follows:

$$q_t = \text{LocalAttn}(q_t + \text{MLP}(\mu_t), q_k + \text{MLP}(g_k)). \tag{6}$$

## 4  Experiments

### 4.1  Experimental Settings

**Dataset and Metrics.** We conduct experiments on the nuScenes dataset [3], a large-scale benchmark specifically curated for autonomous driving research. The dataset comprises 700 training scenes, 150 validation scenes, and 150 test scenes. Each scene includes synchronized sensor data from six surround-view cameras and LiDAR, enabling comprehensive 3D perception across diverse urban environments. Comprehensive annotations are provided to support multiple tasks, including 3D object detection, LiDAR semantic segmentation, and 3D map segmentation. Building upon the nuScenes dataset, SurroundOcc [57] provides the dense 3D semantic occupancy annotation tailored for the occupancy prediction task. The annotated voxel grid spans [-50m, 50m] along both X and Y axes, and [-5m, 3m] along the Z axis with a resolution of $200 \times 200 \times 16$. Each voxel is assigned one of 18 classes, comprising 16 semantic categories, an empty class, and an unknown class.

The quality of semantic occupancy prediction is evaluated using the mean Intersection-over-Union (mIoU) and Intersection-over-Union (IoU) metrics [49]. For 3D object detection, we adopt the standard nuScenes Detection Score (NDS) and mean Average Precision (mAP) metrics [3]. We also contain five true positive (TP) metrics, including ATE, ASE, AOE, AVE, and AAE for measuring translation, scale, orientation, velocity, and attribute errors, respectively.

**Implementation Details.** During the pre-training stage, we adopt a ResNet101-DCN [11] backbone initialized from an FCOS3D [54] checkpoint for the occupancy prediction task, while ResNet50 and ResNet101 backbones that are pre-trained with nuImages [3] for the 3D object detection task. The feature extraction employs a feature pyramid network [25] (FPN), producing multi-scale image representations at downsampling factors of 4, 8, 16, and 32. We configure the Gaussian counts to 25,600, and apply two transformer layers to enhance Gaussian attributes. Model training utilizes the AdamW [35] optimizer, with a 0.01 weight decay. The learning rate linearly warms up over the initial 500 steps to 2e-4 and then follows a cosine decay schedule. Pre-training is conducted for 20 epochs using a batch size of 8. Only random horizontal flipping data augmentation is included. Our implementation is based on MMDetection3D [9]. Fine-tuning follows the official downstream model configurations without modification. All experiments are conducted on a server with 8 GPUs.

Table 1: **3D semantic occupancy prediction results on the SurroundOcc `val` set.** While the original TPVFormer [15] is trained with LiDAR segmentation labels, TPVFormer* is supervised by dense occupancy annotations.

| Method | SC IoU | SSC mIoU | barrier | bicycle | bus | car | const. veh. | motorcycle | pedestrian | traffic cone | trailer | truck | drive. suf. | other flat | sidewalk | terrain | manmade | vegetation |
|---|---|---|---|---|---|---|---|---|---|---|---|---|---|---|---|---|---|---|
| MonoScene [4] | 23.96 | 7.31 | 4.03 | 0.35 | 8.00 | 8.04 | 2.90 | 0.28 | 1.16 | 0.67 | 4.01 | 4.35 | 27.72 | 5.20 | 15.13 | 11.29 | 9.03 | 14.86 |
| Atlas [40] | 28.66 | 15.00 | 10.64 | 5.68 | 19.66 | 24.94 | 8.90 | 8.84 | 6.47 | 3.28 | 10.42 | 16.21 | 34.86 | 15.46 | 21.89 | 20.95 | 11.21 | 20.54 |
| BEVFormer [24] | 30.50 | 16.75 | 14.22 | 6.58 | 23.46 | 28.28 | 8.66 | 10.77 | 6.64 | 4.05 | 11.20 | 17.78 | 37.28 | 18.00 | 22.88 | 22.17 | 13.80 | 22.21 |
| TPVFormer [15] | 11.51 | 11.66 | 16.14 | 7.17 | 22.63 | 17.13 | 8.83 | 11.39 | 10.46 | 8.23 | 9.43 | 17.02 | 8.07 | 13.64 | 13.85 | 10.34 | 4.90 | 7.37 |
| TPVFormer* [15] | 30.86 | 17.10 | 15.96 | 5.31 | 23.86 | 27.32 | 9.79 | 8.74 | 7.09 | 5.20 | 10.97 | 19.22 | 38.87 | 21.25 | 24.26 | 23.15 | 11.73 | 20.81 |
| OccFormer [70] | 31.39 | 19.03 | 18.65 | 10.41 | 23.92 | 30.29 | 10.31 | 14.19 | 13.59 | 10.13 | 12.49 | 20.77 | 38.78 | 19.79 | 24.19 | 22.21 | 13.48 | 21.35 |
| SurroundOcc [57] | 31.49 | 20.30 | 20.59 | 11.68 | 28.06 | 30.86 | 10.70 | 15.14 | 14.09 | 12.06 | 14.38 | 22.26 | 37.29 | 23.70 | 24.49 | 22.77 | 14.89 | 21.86 |
| GaussianFormer [16] | 29.83 | 19.10 | 19.52 | 11.26 | 26.11 | 29.78 | 10.47 | 13.83 | 12.58 | 8.67 | 12.74 | 21.57 | 39.63 | 23.28 | 24.46 | 22.99 | 9.59 | 19.12 |
| **GaussianFormer + SQS (Ours)** | **31.52** | **20.40** | 19.98 | 11.86 | 28.21 | 30.68 | 10.87 | 15.03 | 14.28 | 9.57 | 14.74 | 22.98 | 39.82 | 23.88 | 25.46 | 23.09 | 14.56 | 21.31 |

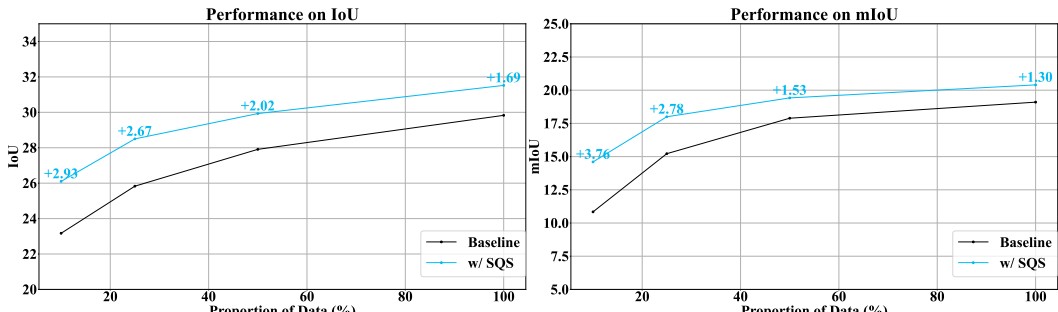

Figure 3: **Data efficiency analysis.** To assess data efficiency under limited annotation scenarios, we reduce the amount of labeled data used for downstream fine-tuning in the 3D semantic occupancy prediction task. The outcomes demonstrate that our pre-training method significantly enhances performance, even when only a small portion of annotations is available.

## 4.2 Main Results

We evaluate the effectiveness of SQS on two challenging downstream perception tasks: semantic occupancy prediction and 3D object detection.

**Semantic Occupancy Prediction.** In Tab. 1, we present a comprehensive quantitative comparison of various methods for multi-view 3D semantic occupancy prediction on the SurroundOcc validation set. Among these methods, GaussianFormer [16] is a novel query-based occupancy prediction method, performing on par with OccFormer [70] and SurroundOcc [57]. After being pre-trained by our method, the GaussianFormer obtains 1.69 IoU and 1.30 mIoU improvements, achieving 20.40% mIoU, when compared to the 19.10% mIoU for GaussianFormer. These results highlight the effectiveness of SQS for the query-based semantic occupancy prediction task.

**3D Object Detection.** We also have conducted experiments in the 3D object detection task, the results are illustrated in Tab. 2. To validate the generality of SQS, we have performed two different sparse object detection methods, e.g., SparseBEV [31] and Sparse4Dv3 [29] on the nuScenes validation set. When leveraging the ResNet50 as the image backbone, and the input image size is $704 \times 256$, SparseBEV achieves 55.8 NDS performance, and an impressive 44.8 mAP metric. After being pre-trained by SQS, we reach the 56.6 NDS and 45.2 mAP performance. Meanwhile, we set the new performance record, that is 56.9 NDS and 47.4 mAP for the Sparse4Dv3 being pre-trained by SQS. Then, we upgrade the backbone to ResNet101 and scale the input size to $1408 \times 512$. Under this setting, the SparseBEV also benefits from our pre-training paradigm with 0.8 mAP and 1.0 NDS improvements. Likewise, Sparse4Dv3 obtains corresponding improvements of 0.7 mAP and 0.8 NDS. The results also validate the effectiveness and generality of our pre-training paradigm.

Table 2: **3D object detection results on the nuScenes** `val` **split.** † benefits from perspective pre-training [31]. ‡ indicates methods with CBGS [73] which will elongate 1 epoch into 4.5 epochs.

| Method | Backbone | Input Size | Epochs | NDS | mAP | mATE | mASE | mAOE | mAVE | mAAE |
|---|---|---|---|---|---|---|---|---|---|---|
| PETRv2 [34] | ResNet50 | 704 × 256 | 60 | 45.6 | 34.9 | 0.700 | 0.275 | 0.580 | 0.437 | 0.187 |
| BEVStereo [21] | ResNet50 | 704 × 256 | 90 ‡ | 50.0 | 37.2 | 0.598 | 0.270 | 0.438 | 0.367 | 0.190 |
| BEVPoolv2 [12] | ResNet50 | 704 × 256 | 90 ‡ | 52.6 | 40.6 | 0.572 | 0.275 | 0.463 | 0.275 | 0.188 |
| SOLOFusion [43] | ResNet50 | 704 × 256 | 90 ‡ | 53.4 | 42.7 | 0.567 | 0.274 | 0.511 | 0.252 | 0.181 |
| Sparse4Dv2 [28] | ResNet50 | 704 × 256 | 100 | 53.9 | 43.9 | 0.598 | 0.270 | 0.475 | 0.282 | 0.179 |
| StreamPETR † [53] | ResNet50 | 704 × 256 | 60 | 55.0 | 45.0 | 0.613 | 0.267 | 0.413 | 0.265 | 0.196 |
| SparseBEV [31] | ResNet50 | 704 × 256 | 36 | 54.5 | 43.2 | 0.606 | 0.274 | 0.387 | 0.251 | 0.186 |
| SparseBEV † [31] | ResNet50 | 704 × 256 | 36 | 55.8 | 44.8 | 0.581 | 0.271 | 0.373 | 0.247 | 0.190 |
| **SparseBEV † + SQS (Ours)** | ResNet50 | 704 × 256 | 36 | 56.6 | 45.2 | 0.564 | 0.263 | 0.362 | 0.232 | 0.182 |
| Sparse4Dv3 † [29] | ResNet50 | 704 × 256 | 100 | 56.1 | 46.9 | 0.553 | 0.274 | 0.476 | 0.227 | 0.200 |
| **Sparse4Dv3 † + SQS (Ours)** | ResNet50 | 704 × 256 | 100 | **56.9** | **47.4** | 0.542 | 0.266 | 0.458 | 0.218 | 0.191 |
| DETR3D † [55] | ResNet101-DCN | 1600 × 900 | 24 | 43.4 | 34.9 | 0.716 | 0.268 | 0.379 | 0.842 | 0.200 |
| BEVFormer † [24] | ResNet101-DCN | 1600 × 900 | 24 | 51.7 | 41.6 | 0.673 | 0.274 | 0.372 | 0.394 | 0.198 |
| BEVDepth [22] | ResNet101 | 1408 × 512 | 90 ‡ | 53.5 | 41.2 | 0.565 | 0.266 | 0.358 | 0.331 | 0.190 |
| Sparse4D † [26] | ResNet101-DCN | 1600 × 900 | 48 | 55.0 | 44.4 | 0.603 | 0.276 | 0.360 | 0.309 | 0.178 |
| SOLOFusion [43] | ResNet101 | 1408 × 512 | 90 ‡ | 58.2 | 48.3 | 0.503 | 0.264 | 0.381 | 0.246 | 0.207 |
| SparseBEV † [31] | ResNet101 | 1408 × 512 | 24 | 59.2 | 50.1 | 0.562 | 0.265 | 0.321 | 0.243 | 0.195 |
| **SparseBEV † + SQS (Ours)** | ResNet101 | 1408 × 512 | 24 | 60.2 | 50.9 | 0.531 | 0.251 | 0.318 | 0.241 | 0.185 |
| Sparse4Dv3 † [29] | ResNet101 | 1408 × 512 | 100 | 62.3 | 53.7 | 0.511 | 0.255 | 0.306 | 0.194 | 0.192 |
| **Sparse4Dv3 † + SQS (Ours)** | ResNet101 | 1408 × 512 | 100 | **63.1** | **54.4** | 0.498 | 0.241 | 0.298 | 0.187 | 0.188 |

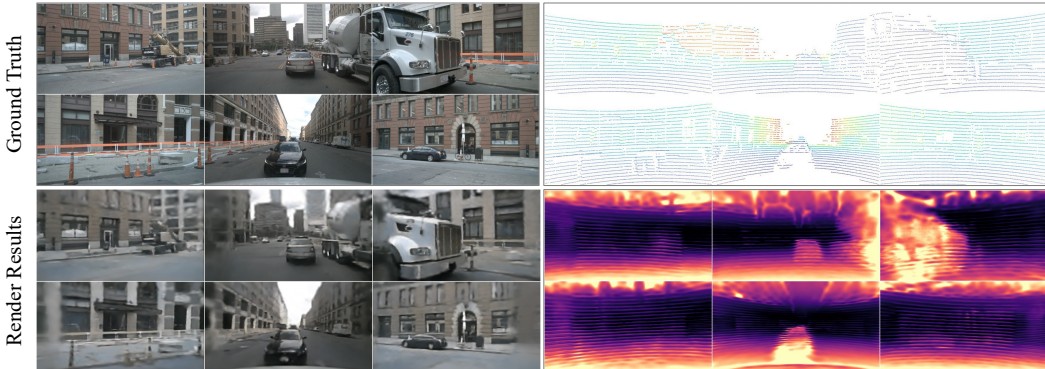

Figure 4: **Rendering results visualization.** Leveraging multi-view images and depth maps projected by the sparse point cloud as supervision, SQS demonstrates compelling depth and image reconstruction after pre-training.

**Data Efficiency.** One of the main advantages of pre-training lies in its ability to improve data efficiency for downstream tasks, especially when annotated data is limited. To further demonstrate the effectiveness of our pre-training strategy under conditions where there is plenty of pre-training data but restricted access to labeled downstream samples, we fine-tune our model—initially pre-trained on the full dataset—using different fractions (10%, 25%, 50%, and 100%) of the SurroundOcc training set.

Fig. 3 showcases how SQS improves data efficiency. With full fine-tuning data, SQS yields improvements of +1.69 IoU and +1.3 mIoU over the baseline. Notably, this benefit is amplified as less fine-tuning data is used: for instance, fine-tuning with only 10% of the data results in a gain of about +3.7 mIoU. These findings highlight the strength of SQS in achieving notable performance improvements through query-based splatting pre-training, particularly when downstream annotated data is scarce.

**Visualization of Renderings.** As shown in Fig. 4, employing 25,600 queries for 3DGS reconstruction through the multi-view RGB images and depth maps as supervision, SQS could predict promising depth and RGB images during the pre-training stage.

Table 3: **Ablation studies.** We report the IoU and mIoU metrics on the SurroundOcc *val* set for the 3D semantic occupancy prediction task. "Rend.", "B.b." and "Inter." denote rendering, image backbone, and query interaction, respectively.

| Methods | Rend. RGB | Rend. Depth | Load B.b. | Query Inter. | IoU | mIoU |
|---|---|---|---|---|---|---|
| Baseline [16] | | | | | 25.8 | 15.2 |
| Model A | ✓ | | ✓ | | $23.8^{\downarrow 2.0}$ | $12.2^{\downarrow 3.0}$ |
| Model B | | ✓ | ✓ | | $27.9^{\uparrow 2.1}$ | $17.3^{\uparrow 2.1}$ |
| Model C | ✓ | ✓ | ✓ | | $28.2^{\uparrow 2.4}$ | $17.5^{\uparrow 2.3}$ |
| Model D | ✓ | ✓ | | ✓ | $26.3^{\uparrow 0.5}$ | $15.9^{\uparrow 0.7}$ |
| Model E | | | | ✓ | $25.7^{\downarrow 0.1}$ | $15.3^{\uparrow 0.1}$ |
| **SQS (Ours)** | ✓ | ✓ | ✓ | ✓ | $\mathbf{28.5}^{\uparrow 2.7}$ | $\mathbf{18.0}^{\uparrow 2.8}$ |

## 4.3 Ablation Studies

In this section, we conduct ablations on the semantic occupancy prediction task on the validation split of the SurroundOcc dataset. In order to reduce the training time, we utilize the quarter of training data during the pre-training and fine-tune stage for all experiments. The results are demonstrated in Tab. 3.

**Rendering Objectives.** We first investigate the impact of various rendering objectives during the pre-training stage. Specifically, in ModelA, ModelB, and ModelC of Tab. 3, we employ RGB rendering only, depth rendering only, and a combination of both RGB and depth rendering as the pre-training objectives, respectively. The results reveal that utilizing only RGB rendering during pre-training impairs fine-tuning performance, resulting in a reduction of 2.0 in IoU and 3.0 in mIoU. In contrast, incorporating depth rendering alone leads to improvements of 2.1 in both IoU and mIoU metrics. These findings suggest that rendered depth supervision enhances the geometric representation capability of the pre-trained model, thereby facilitating improved fine-tuning performance. Furthermore, when both RGB and depth renderings are jointly applied, we observe a marginal additional improvement. This indicates that rendered RGB supervision provides supplementary benefits in the presence of rendered depth supervision.

**Effects of Query Interaction.** To further assess the impact of query interaction during the fine-tuning stage, we develop Model D, which exclusively incorporates the query interaction mechanism during fine-tuning. As presented in Tab. 3, the pre-trained model is capable of generating meaningful queries for reconstruction, which can be further leveraged to enhance the query learning process through the query interaction module during fine-tuning. This results in improvements of 0.5 IoU and 0.7 mIoU. To eliminate the influence of extra query interaction during the fine-tuning stage, we additionally design Model E to exclusively adapt the query interaction module without pre-training. Its performance remains nearly identical to that of the baseline, indicating that the additional query interaction module offers negligible benefit during fine-tuning. Finally, by initializing with the pre-trained image backbone and FPN neck, we obtain optimal fine-tuning performance, reaching 28.5% IoU and 18.0% mIoU. These results demonstrate the superiority of the query interaction design within our proposed SQS paradigm.

## 4.4 Limitations and Future Work

While SQS has achieved further improvements across various downstream tasks, becoming a plug-and-play general pre-training paradigm for sparse perception models, it still faces several limitations. One limitation is the extra computation burden and memory consumption incurred by the plug-in pre-training model. Another limitation is the insufficient utilization of pre-training queries for different downstream tasks.

In the future, we will explore how to introduce the semantic information during the pre-training stage and then use the semantic information to distinguish the pre-trained queries for various downstream tasks. We will also try to apply the SQS to query-based end-to-end autonomous driving approaches such as SparseAD [67] and GaussianAD [71].

## 5 Conclusion

In this paper, we introduced SQS, a novel query-based splatting pre-training paradigm tailored for autonomous driving SPMs. SQS overcomes the limitations of previous pre-training methods by enabling image backbone and Gaussian queries to learn rich 3D representations through 3D Gaussian prediction and the reconstruction of both images and depth maps. The plug-in design and query interaction strategy further allow seamless transfer and adaptation of the pre-trained model to diverse downstream tasks. Extensive experiments on benchmark datasets validate the effectiveness of SQS, showing promising improvements over various SOTA SPMs.

## Acknowledgements

This work was supported by NSFC with Grant No. 62573371, by the Basic Research Project No. HZQB-KCZYZ-2021067 of Hetao Shenzhen-HK S&T Cooperation Zone, by Guangdong S&T Programme with Grant No. 2024B0101030002, by the Shenzhen General Program No. JCYJ20220530143600001, by the Shenzhen Outstanding Talents Training Fund 202002, by the Guangdong Research Project No.2017ZT07X152 and No. 2019CX01X104, by the Guangdong Provincial Key Laboratory of Future Networks of Intelligence (Grant No. 2022B1212010001), by the Guangdong Provincial Key Laboratory of BigData Computing CHUK-Shenzhen, by the NSFC 61931024&12326610&62293482, by the Key Area R&D Program of Guangdong Province with grant No. 2018B030338001, by the Shenzhen Key Laboratory of Big Data and Artificial Intelligence (Grant No. SYSPG20241211173853027), by China Association for Science and Technology Youth Care Program, by the Shenzhen-Hong Kong Joint Funding No. SGDX20211123112401002, and by Tencent & Huawei Open Fund.

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
