# OpenReview forum: "SQS: Enhancing Sparse Perception Models via Query-based Splatting in Autonomous Driving"
_NeurIPS.cc/2025/Conference — NeurIPS 2025 spotlight_

### Official Review · Reviewer_zzoE · 2025-06-22

**Clarity:** 3
**Significance:** 2
**Originality:** 2
**Rating:** 4
**Confidence:** 3

**Summary:**

This paper proposes SQS, a pre-training framework for Sparse Perception Models (SPMs) in autonomous driving. SQS introduces query-based 3D Gaussian splatting to enable efficient and effective scene reconstruction from multi-view images and depth maps. The method leverages self-supervised learning during pre-training to capture fine-grained contextual features, which are then transferred to downstream tasks via an interactive query fusion mechanism during fine-tuning. Experimental results demonstrate that SQS improves performance on 3D semantic occupancy prediction and object detection tasks compared to previous state-of-the-art approaches.

**Questions:**

1) What is the core technical novelty of SQS beyond combining existing components (3DGS and query-based architectures) into a self-supervised pre-training framework?

2) Can the authors provide evidence of cross-dataset generalization of the proposed pre-training method?

3) Can the authors provide SOTA methods in the comparison?

**Ethical Concerns:**

["NO or VERY MINOR ethics concerns only"]

**Final Justification:**

The authors have addressed my concerns, and I believe the paper meets the acceptance bar; however, I hope the revised version includes the promised discussions and experiments to further strengthen the work.

**Limitations:**

Yes

**Quality:**

2

**Strengths And Weaknesses:**

Strengths:
1) The experimental evaluation includes two key tasks: semantic occupancy prediction and 3D object detection. Comparisons with state-of-the-art methods demonstrate consistent performance improvements.

2) The paper is well organized and easy to follow.

Weaknesses:
1) The overall novelty of the paper is somewhat limited. While the proposed method, SQS, introduces a query-based pre-training framework for Sparse Perception Models (SPMs) using 3D Gaussian Splatting (3DGS), both 3DGS and query-based architectures are well-established techniques that have been explored in prior works. The core contribution of the paper lies more in extending these existing components into the domain of self-supervised pre-training for autonomous driving, rather than proposing fundamentally new technical innovations.

2) the evaluation of the pre-training effectiveness is primarily conducted on a single dataset (nuScenes), albeit across multiple downstream tasks. This approach may not fully demonstrate the generalization capability of the proposed pre-training paradigm. Given that different datasets can vary significantly in terms of LiDAR configuration, camera ISP characteristics, and the accuracy of pose annotations—factors that can greatly influence the quality of rendered RGB and depth images—the performance of a pre-training method should ideally be validated across multiple diverse datasets. A more comprehensive cross-dataset evaluation would strengthen the claim of the method’s robustness and broader applicability.

3) The paper does not compare against all possible recent baselines (both in 3D det and 3D Occ), which slightly weakens the strength of the comparison.

4) The description of the query interaction mechanism during fine-tuning (Section 3.5) is somewhat dense and could be improved with a diagram or step-by-step explanation to make it more accessible.

---

> ### Author Rebuttal · Authors · 2025-07-31
>
> We thank you for the careful evaluation of our work and the valuable feedback. We address each comment in detail below:
>
> **1. The Core Technical Novelty**
>
> Thank you for your feedback and for highlighting the scope of our contributions. We acknowledge that 3D Gaussian Splatting and query-based architectures are established techniques. However, as the reviewers **DKKC** and **uHvD** concluded, our primary focus is on the novel integration of these components within a self-supervised pre-training framework specifically tailored for Sparse Perception Models (SPMs) in autonomous driving. The plug-and-play Gaussian queries can be conveniently integrated within SPMs using the proposed Query Interaction mechanism, leading to improved downstream performance without reliance on extensive labeled data.
>
> We hope to clarify the novelty of our approach in the revised manuscript and appreciate your constructive comments that help us improve the presentation of our work.
>
>
> **2. Cross-dataset Generalization Analysis**
>
> We appreciate your insightful comment regarding the cross-dataset evaluation of our method. While our current study primarily focuses on the nuScenes dataset, we selected it due to its comprehensive sensor setup and rich annotations. We agree that evaluating across datasets would show the generalization capabilities of our approach.
>
> To further evaluate, we conduct experiments on the KITTI-360[1] dataset for 3D semantic occupancy prediction with monocular camera, which differs significantly in sensor setup compared with nuScenes. GaussianFormer is selected as the Sparse Perception Model (SPM) to be improved with our SQS. We use the dense semantic occupancy annotations from SSCBench-KITTI-360[2] for supervision and evaluation. We utilize all training data for pre-training and only **half** data for fine-tuning due to the time limitation.
>
> After being integrated with our method, the GaussianFormer obtains **0.48** mIoU and **1.74** IoU improvements, achieving **7.70\%** mIoU and **29.70\%** IoU, when compared to the **7.22\%** mIoU and **27.96\%**  IoU without SQS. The results successfully demonstrate the generalization of our SQS. Please note that the results could be further improved through hyperparameter tuning, provided that sufficient time is available for experimentation. We will update these results in our revised version of the paper.
>
> **3. Compare with SOTA Methods**
>
> Thanks for your insightful comment. We agree that comprehensive comparisons are important. For the 3D semantic occupancy prediction, we evaluate SQS using the SurroundOcc benchmark, and all possible recent baselines in this dataset are included for comparison. For the 3D object detection, we primarily choose SparseBEV as the baseline, given its balanced performance and model complexity. In the revised version, we plan to include results from Sparse4Dv3 and other more recent 3D detection methods in the table.
>
> Importantly, through comparison, we aim to demonstrate that using SQS can significantly improve the performance of current popular SOTA SPMs, rather than solely striving for SOTA metrics in 3D Occupancy and Detection tasks.
>
> **4. More Accessible Explanation for The Query Interaction Mechanism**
>
> Thank you for your valuable feedback. We acknowledge that the description of the query interaction mechanism is complex and may benefit from additional clarification. To address this, we will include a diagram illustrating the process flow and step-by-step explanations to enhance clarity in the revised version. The step-by-step explanations are listed as follows:
>
> 1. Infer the pre-trained Gaussian queries paired with anchors using the fixed pre-trained model;
>
> 2. Obtain Gaussian queries from step 1 by filtering out corresponding anchors with low opacity;
>
> 3. In the downstream SPMs, each task queries only aggregate features from $k$ closest 3D Gaussians selected from step 2 by applying $k$-nearest neighbor algorithm.
>
> [1] Yiyi Liao, Jun Xie, and Andreas Geiger. Kitti-360: A novel dataset and benchmarks for urban scene understanding in 2d and 3d. IEEE Transactions on Pattern Analysis and Machine Intelligence, 45(3):3292–3310, 2022.
>
> [2] Yiming Li, Sihang Li, Xinhao Liu, Moonjun Gong, Kenan Li, Nuo Chen, Zijun Wang, Zhiheng Li, Tao Jiang, Fisher Yu, et al. Sscbench: A large-scale 3d semantic scene completion benchmark for autonomous driving. In 2024 IEEE/RSJ International Conference on Intelligent Robots and Systems (IROS), pages 13333–13340. IEEE, 2024.

---

> > ### Comment · Reviewer_zzoE · 2025-08-02
> >
> > Thank you for your response. I have carefully reviewed it and am satisfied that my concerns have been adequately addressed. I appreciate the additional clarifications and hope the rebuttal discussions are reflected in the final version.

---

> > > ### Author Response · Authors · 2025-08-07
> > >
> > > Thank you for your feedback and the update! We're pleased that your primary concerns have been resolved and will make sure to include the additional analyses in the final version.

---

### Official Review · Reviewer_aTUF · 2025-06-29

**Clarity:** 3
**Significance:** 3
**Originality:** 3
**Rating:** 4
**Confidence:** 2

**Summary:**

This paper proposes a method to a plug-in module that predicts 3D Gaussian representations
from sparse queries during pre-training and uses the pretrained 3D Gaussian queries to fien-tune the downstream tasks.

**Questions:**

The reviewer acknowledges that the proposed method do have improvement (vital or subtle across different object categories) with utilization of GaussianFormer + the proposed un-supervised (semi-supervised method with the depth image utlized?). This experiment has demonstrated that SQS can improve the performance of GaussianFormer. However, the paper has claimed that the SQS is somehow model agnostic. Therefore, the reviewer do recommend the paper to add experiments that combine SQS to other models (maybe not the SOTA one, not necessary) to see if SQS has a stable improvement across different backbone models. By adding such an experiment, it can demonstrate the generality of SQS, possibly increase the solidity.

**Ethical Concerns:**

["NO or VERY MINOR ethics concerns only"]

**Final Justification:**

The main concern of the reviewer is solved. Please update the content in rebuttal to the paper.

**Limitations:**

It includes the limitation of resource costing of the proposed method.

**Paper Formatting Concerns:**

None.

**Quality:**

3

**Strengths And Weaknesses:**

S1. It proposes a method which leverage un-annotated data, which is promissing.

S2. The Sparse Query utilization seems novel.

S3. The improvement of object detection seems works on all object categories.

W1. The clarity of the presentation should be revised. The reviewer has difficulty on understanding the training of sparse queries and how they are used in the fine-tuning procedure (even though there is Figure 2).

W2. Experiment can be insufficient.

---

> ### Author Rebuttal · Authors · 2025-07-31
>
> We thank you for the careful evaluation of our work and the valuable feedback. We address each comment in detail below:
>
> **1. Clarity on Query Training and Fine-tuning Procedure with Figure 2**
>
> We appreciate the reviewer’s concern regarding the clarity of our description about sparse query training and their role during fine-tuning. We recognize that additional textual explanation could complement Figure 2 and improve understanding.
>
> To clarify, our proposed SQS introduces a plug-in module that predicts 3D Gaussian representations from sparse queries via a Gaussian Transformer Decoder in the top row of Figure 2 during pre-training, leveraging self-supervised splatting to learn fine-grained contextual features through the reconstruction of multi-view images and depth maps. In the downstream fine-tuning stage, we could share the pre-trained image encoder weights for downstream model initialization, and the pre-trained Gaussian queries are seamlessly integrated into downstream networks via query interaction in the Transformer Decoder of sparse perception models (SPMs) to enhance the performance.
>
> In the revised version, we will further elaborate on the query interaction pipeline, supplement Figure 2 with additional explanatory notes, and provide a step-by-step procedure in the main text for greater clarity.
>
> **2. Generality of SQS Across Backbone Models**
>
> Thank you for pointing out the importance of thoroughly demonstrating the model-agnostic claim of SQS. As you observed, our current experiments focus on integrating SQS with GaussianFormer for occupancy prediction and SparseBEV for the 3D object detection task. And in Table 2, we have conducted experiments on two different image backbones of SparseBEV to validate the effectiveness of our method.
>
> To further evaluate the generality of SQS, we have performed additional experiments, integrating SQS into another sparse perception model e.g., Sparse4Dv3, as pointed by Reviewer uHvD. On the nuScenes validation set, under limited training (using only 1/4 of the training data to accelerate training), Sparse4Dv3 achieves **29.5** mAP and **36.1** NDS. Incorporating our SQS pre-training module leads to further improvements, reaching **30.4** mAP and **37.3** NDS. These new results combined with the results of SparseBEV could further validate the generality of SQS across different backbone models.
> The final full dataset results will be detailed in our final version.
>
> These results validate the model-agnostic property of the SQS framework and demonstrate its stable performance enhancement across architectures. We greatly appreciate the reviewer’s suggestion and agree that this substantially increases the solidity and generalizability of our method.

---

> > ### Comment · Reviewer_aTUF · 2025-08-03
> >
> > The reviewer thanks the response of the authors. Please update the added experiment results into the main content.

---

> ### Author Response · Authors · 2025-08-07
>
> Thank you for the positive comments and the update! We’re pleased to know that your main concerns have been resolved and we will include the additional analyses in the final version.

---

### Official Review · Reviewer_uHvD · 2025-07-01

**Clarity:** 4
**Significance:** 2
**Originality:** 3
**Rating:** 5
**Confidence:** 3

**Summary:**

This paper presents SQS, a pretraining framework for sparse 3D perception models targeted towards improving downstream performance in perception tasks for autonomous driving vehicles. Unlike existing dense BeV-centric pretraining methods, SQS introduces adaptive Gaussian queries and a query splatting mechanism that uses a self-supervised learning approach to learn meaningful representations.

In this pretraining, SQS reconstructs RGB and depth from multi-view inputs and produces 3D Gaussian queries. These learned queries can then be used in downstream tasks (occupancy prediction, 3D object detection) using a query interaction module.

**Questions:**

Q1. I would like the authors to explain why Sparse4Dv3 is not included as a baseline in Table 2 (3D detection on nuScenes). In lieu of such an explanation, can the authors explain why they believe it is not necessary to achieve SOTA performance for this proposed method to be valuable, at least in relation to 3D detection? Perhaps there are other advantages, e.g., latency, memory, etc.? While not achieving SOTA in and of itself is not *necessarily* a weakness, the method must then be motivated in other ways. If this point cannot be justified, we are left with SQS allowing for SOTA performance in just one downstream perception task.

Q2. Let's ignore W1 about not beating Sparse4Dv3 for now, and this is not necessarily a weakness -- Experiments are performed on two different downstream perception tasks. Showing that this pretraining framework improves performance in even more tasks would show that this approach could in fact be scaled up to production-level autonomous driving stacks. Did the authors experiment with any other tasks? If so, what did the results look like? If not, are there barriers to applying this to other tasks? Or are there plans for future work? Again, two downstream tasks is sufficient in my view for a single conference paper, and this is a question moreso out of curiosity.

Q3. See W3 -- Can the authors provide analysis on how query reuse transpires when deployed? Either quantitatively (how often are queries reused, when are they reused, are there certain factors with which reuse is a covariate?) or with visualizations (is there a clever way we can visualize what the queries are actually learning, and how they may be reused?)

Q4. Can the authors provide more analysis about the real-world applicability (i.e., latency and memory) of this pretraining method, as compared to existing works?

**Ethical Concerns:**

["NO or VERY MINOR ethics concerns only"]

**Final Justification:**

The rebuttal has addressed my few concerns about the paper, so I am raising my score to an acceptance.

**Limitations:**

Yes.

**Paper Formatting Concerns:**

None.

**Quality:**

3

**Strengths And Weaknesses:**

S1. The introduction of a novel sparse query-based splatting pretraining framework seems to be novel and is well-motivated by and aligned with the sparse query paradigm in recent perception systems. Ablation studies show that removing components of the method design does indeed degrade performance.

S2. This pertraining method can be seen as a versatile plug-and-play framework for different downstream perception tasks, demonstrated by competent performance in two different such tasks: occupancy prediction and 3D object detection. This suggests that this framework *could* generalize to training larger more complete production-level autonomous driving systems.

S3. The presentation is coherent, the method pipeline figure is simple and clean, and the visualizations are supportive of the claims made in the paper.

W1. Lack of comparison with Sparse4Dv3 (2023). This is a significant omission in Table 2 (3D object detection on nuScenes), as it seems to me that Sparse4Dv3 is a relevant baseline that outperforms the presented method across many metrics, including NDS and mAP. For that matter, Sparse4Dv2 seems to also beat the presented method in some metrics. I would like the authors to argue why beating Sparse4Dv3 on this task is not necessary. Without a clear explanation, this reduces the number of tasks that this pretraining framework achieves SOTA performance in to just one. This may undercut the Significance of the proposed method.

W2. Latency/memory comparisons are not made against baseline methods. Including this and an analysis would greatly improve the real-world relevance of this proposed pretraining framework.

W3. Lack of in-depth study about success/failure modes. The paper discusses reuse of queries across tasks, but a study of this is not shown. Is there any way to quantify or visualize this supposed reuse of queries, and how it impacts performance in specific cases, whether in a positive or negative manner?

---

> ### Author Rebuttal · Authors · 2025-07-31
>
> We sincerely thank the reviewer for the detailed feedback and the opportunity to clarify and strengthen our work. We address each point below:
>
> **1. Comparison with Sparse4Dv3 Performance**
>
> We appreciate the reviewer’s concern about the lack of comparison to Sparse4Dv3. Our initial choice of SparseBEV as the primary baseline was motivated by its balanced design in terms of detection accuracy and model complexity, and because sparse detection frameworks generally share similar architectural paradigms. While the Sparse4D series has indeed introduced innovations in temporal fusion, many core mechanisms of sparse detection remain comparable across these approaches, making SparseBEV a widely recognized benchmark.
>
> To further enhance the rigor of our work and address this omission, we have conducted additional experiments incorporating Sparse4Dv3. On the nuScenes validation set, under limited training (using only 1/4 of the training data), Sparse4Dv3 achieves **29.5** mAP and **36.1** NDS, while incorporating our SQS pre-training module leads to improvements, reaching **30.4** mAP and **37.3** NDS. These results demonstrate that our pre-training framework also benefits more advanced architectures like Sparse4Dv3, and thus our findings are not specific to a single backbone or detection framework. We will update the final version of our paper with full-scale results.
>
> **2. Latency and Memory Analysis**
>
> Thank you for drawing attention to the question of real-world applicability. We recognize the importance of benchmarking computation and memory efficiency. Our SQS framework is designed with plug-and-play considerations: the extra computational overhead is predominantly introduced during pre-training via the query interaction module. During downstream fine-tuning and inference, it is possible to selectively activate or entirely omit the query interaction module. For most deployment scenarios, loading only the pre-trained image encoder introduces negligible additional latency or memory consumption compared to vanilla baselines, while still reaping the representational benefits of pretraining.
>
> We have conducted preliminary latency and memory profiling with different variants of the SQS modules across the occupancy prediction baseline (i.e., GaussianFormer) as responded to Reviewer DKKC, and observed that:
> The SQS-pretrained image encoder adds no extra inference latency when the query interaction module is omitted at deployment.
> When using the partial module sharing strategy, the latency increases by approximately **30\%**, and GPU memory consumption increases by **35\%** compared to baseline, with a notable performance boost on downstream tasks.
> These findings suggest that our method is not only flexible, but also practical for scalable real-world applications. We will provide detailed results and hardware configurations in the final version of our paper.
>
> **3. Study and Visualization of Query Reuse Across Tasks**
>
> We appreciate the reviewer’s insightful question regarding the quantitative analysis and visualization of query reuse across tasks, as well as how such reuse impacts downstream performance.
>
> In our current implementation, the query interaction mechanism adopts a purely adaptive self-attention framework, where all pre-trained queries are dynamically weighted and selected according to the learned attention scores. This design enables flexible transfer of information but does not explicitly label or fix certain queries for reuse in particular downstream tasks. As such, it is inherently challenging to unambiguously trace and quantify “reused queries” or directly visualize their task-level roles, since query participation is emergent and distributed rather than discrete or statically assigned.
>
> We acknowledge this limitation in our analysis. At the same time, we believe this adaptivity is a strength: self-attention enables each downstream task module to discover and leverage the most relevant subset of features, rather than hard-coding query-task assignments during pre-training. Consequently, improvements in downstream performance—in both detection and occupancy prediction—serve as indirect evidence that beneficial query reuse and knowledge transfer are indeed taking place.
>
> For future work, we plan to conduct more comprehensive interpretability studies. For instance, techniques such as analyzing the learned attention distributions, performing query ablation, and visualizing per-query activation maps in downstream heads will likely yield more direct insights into how queries are reused or specialized for specific tasks. We see this as a promising direction for ongoing and follow-up research, and we are grateful to the reviewer for this suggestion.

---

### Official Review · Reviewer_DKKC · 2025-07-03

**Clarity:** 3
**Significance:** 4
**Originality:** 4
**Rating:** 5
**Confidence:** 4

**Summary:**

The paper titled "SQS: Enhancing Sparse Perception Models via Query-based Splatting in Autonomous Driving" introduces a novel pre-training technique called SQS (Sparse Query-based Splatting). It is designed to improve the performance of SPMs (Sparse Perception Models) in autonomous driving. These models are often used for their efficiency and inference speed, brought by the sparse nature of their representations, unlike BEVs or volumetric representations. The SQS method uses a module which predicts 3D Gaussian representations from sparse queries during pre-training, relying on self-supervised splatting to learn contextual features through reconstruction. During fine-tuning, pre-trained Gaussian queries are used in the downstream networks through queries. This connects pre-trained queries with task-specific queries. This makes the method usable for different tasks.
Evaluations show significant improvements over the state of the art on both 3D object detection and occupancy prediction.  Authors discuss limitations of SQS such as the additional computational and memory loads due to the plug-in pre-training model. Authors also propose paths to explore for future works. The paper also discusses the limitations of SQS, including the additional computational burden.

**Questions:**

Authors mention potential computational overhead. Could they discuss potential solutions to mitigate this issue and make their contribution even more valuable?

Since authors mention that pre-training queries are not fully utilized for some downstream tasks, could they discuss which challenges this raises and if they envision any approach which could improve this issue?


The paper mentions the introduction of semantic information during pre-training as a future direction. Could the authors elaborate on how this could be done and how this would impact thbe pre-training?

**Ethical Concerns:**

["NO or VERY MINOR ethics concerns only"]

**Final Justification:**

This work is a solid contribution and authors provided clear and very detailed answers to my remarks, notably on computational overhead. Other reviewers also seem to agree that this work could be suitable for this conference. Therefore I will not update my rating and leave it at 5.

**Limitations:**

yes

**Quality:**

3

**Strengths And Weaknesses:**

## Strengths
The proposed approach SQS is a novel contribution which can be very valuable to the field of autonomous driving,. The approach addresses limitations of existing pre-training methods and offers performance improvements. The self-superviseed nature of the splatting to learn contextual features is also very interesting as it reduces reliance on labeled data, which is sometimes scarce. The integration of the pre-trained Gaussian queries into downstream networks is also a strong idea.
The paper is clear, well-structured, and readable. Figures help to understand the overall architecture and method, and are clear and of good quality. Supplementary materials provide more experimental results and a discussion on metrics, which are both very welcome.

## Weaknesses
Table 1 is a bit small which makes it less readable, especially considering the number of figures. In Figure 4, the top right illustration is difficult to see. Maybe try a different background or color mapping. Authors acknowledge that the pre-training queries are not fully utilized for different downstream tasks. This means there might be some room for improvement in the design of the query interaction mechanisms. Authors could discuss potential improvements.

---

> ### Author Rebuttal · Authors · 2025-07-31
>
> We sincerely thank you for your careful evaluation and constructive feedback. We address each specific concern below:
>
> **1. Table and Figure Readability (Table 1, Figure 4):**
>
> Thank you for highlighting the readability issues. In the final version, we will enlarge Table 1 and reformat it to improve its clarity and readability. For Figure 4, we will adjust the color mapping and background of the top-right figure, to enhance visual contrast and ensure details are more easily discernible.
>
> **2. Fully Utilization of Pre-training Queries and Challenge Discussion:**
>
> We mentioned that the current design does not fully exploit the pre-training queries for different downstream tasks in the limitation section. As described in Sec. 3.3 of our main paper, SQS leverages a self-supervised splatting module to extract contextual 3D Gaussian representations from sparse queries for reconstructing the RGB images and depth maps of the **whole scenario** in the pre-training stage. However, even though our SQS is a generic plug-in method for sparse perception models (SPMs), the role of these queries diverges depending on the downstream applications. For example, in occupancy prediction, queries represent both foreground and background, while in object detection, only foreground queries are learned. While in SQS, the query interaction module is utilized to learn from pre-trained queries **adaptively** without considering this discrepancy, achieving promising performance improvements.
>
> A potential direction to address this is to introduce explicit semantic information (e.g., pseudo-labels or clustering-based semantic grouping) during pre-training to annotate queries according to their semantic categories. This would enable more adaptive and targeted query interaction in downstream tasks, allowing the selection or weighting of queries based on semantic relevance. Such approaches could facilitate more effective knowledge transfer and potentially enhance performance, particularly in tasks requiring fine-grained discrimination.
>
> **3. Incorporating Semantic Information in Pre-training:**
>
> Following the above, incorporating semantic information during pre-training can enhance the query representation’s semantic awareness. This could be achieved by associating each query with semantic pseudo-labels, derived from clustering on features or through auxiliary losses related to semantic segmentation or object proposals. Introducing this semantic distinction during pre-training would allow queries to encode more discriminative and task-relevant features, boosting their transferability and downstream task adaptability. For instance, in the object detection head, only queries pre-trained with ‘foreground’ semantics would participate in the detection decoder, leading to improved sample efficiency and interpretability.
>
> **4. Computational Overhead and Possible Solutions:**
>
> We acknowledge the concern about additional computational overhead and memory consumption, mainly due to the plug-in mechanism introduced by SQS, as illustrated in the table below for the occupancy prediction task with the quarter of training data during the pre-training and fine-tune stage. To address this, we propose several feasible strategies:
>
> - **Tight Computation Application:** In scenarios with tight computational constraints, users can opt to load only the pre-trained image encoder (backbone and neck) and omit the query interaction module during fine-tuning (SQS* in table below). This setup incurs no extra overhead compared to conventional models while still benefiting from the representational improvements gained through pre-training.
>
> - **Partial Module Sharing:** It is also feasible to share the image encoder between the Gaussian queries (for query interaction) and downstream task queries, only enabling the Gaussian Transformer Decoder during fine-tuning. As detailed in the table below for SQS**, this results in a modest overhead but provides a favorable accuracy-resource trade-off.
>
> - **Future Optimization:** We are actively exploring model compression and distillation techniques that can further reduce memory consumption and computation, without sacrificing performance.
>
> | Methods                | IoU  | mIoU | Latency | Memory  |
> |------------------------|------|------|---------|---------|
> | GaussianFormer         | 25.8 | 15.2 | 350ms   | 6100 M  |
> | GaussianFormer + SQS   | 28.5 | 18.0 | 560ms (↑60%)   | 10880 M (↑78%) |
> | GaussianFormer + SQS*  | 28.2 | 17.5 | 350ms (↑0%)   | 6100 M (↑0%) |
> | GaussianFormer + SQS** | 28.3 | 17.8 | 452ms (↑30%)   | 8250 M (↑35%) |
>
> We hope these clarifications directly address your questions and demonstrate our commitment to maximizing both scientific contribution and practical accessibility.

---

> > ### Comment · Reviewer_DKKC · 2025-08-05
> >
> > The authors have written a clear and thorough rebuttal and addressed my concerns. They promised clear solutions to improve readability and gave thoughts to propose strategies to mitigate computational overheads, which I especially appreciated. Overall, the paper presents a strong contribution to the field.

---

> > > ### Author Response · Authors · 2025-08-07
> > >
> > > Thank you for the update and positive feedback! We’re glad to hear your major concerns have been addressed and will incorporate the added analyses into the final version.

---

### Decision · Program_Chairs · 2025-09-17

**Decision:**

Accept (spotlight)

**Comment:**

This paper introduces a novel pre-training approach for sparse perception models in self-driving. The proposed approach is based on predicting Gaussian-splatting representations. Results show improved accuracy and greater data efficiency.
Reviewers all agree on acceptance, pointing to the novelty of the ideas as well as the versatility of the approach in terms of the downstream tasks it improves.